# DiVE-k: differential visual reasoning for fine-grained image recognition

**Raja Kumar[1], Arka Sadhu[2], Ram Nevatia[1]**
[1]University of Southern California, [2]Meta
kumarraj@usc.edu & nevatia@usc.edu
https://raja-kumar.github.io/projects/dive-k.html

## Abstract

Large Vision Language Models (LVLMs) possess extensive text knowledge but struggle to utilize this knowledge for fine-grained image recognition, often failing to differentiate between visually similar categories. Existing fine-tuning methods using Reinforcement Learning (RL) with exact string-match reward are often brittle, encourage memorization of training categories, and fail to elicit differential reasoning needed for generalization to unseen classes. To address this, we propose **DiVE-k**, **Di**fferential **V**isual r**E**asoning using top-**k** generations, framework that leverages model's own top-k predictions as a training signal. For each training image, DiVE-k creates a multiple-choice question from the model's top-k outputs and uses RL to train the model to select the correct answer. This approach requires the model to perform fine-grained differential reasoning among plausible options and provides a simple, verifiable reward signal that mitigates memorization and improves generalization. Experiments on five standard fine-grained datasets show that our method significantly outperforms existing approaches. In the standard base-to-novel generalization setting, DiVE-k surpasses the QWEN2.5-VL-7B and ViRFT by $10.04\%$ and $6.16\%$ on the Harmonic Mean metric, respectively. Further experiments show similar gains in mixed-domain and few-shot scenarios.

## 1 Introduction

We explore the task of zero-shot fine-grained image recognition, as a visual reasoning task building on available Large Vision Language Models (LVLMs). Such capabilities are crucial for the generalization of vision systems. In early zero-shot image recognition works, such as in CLIP (Radford et al., 2021), the visual embedding from an image is matched against the text embedding of class names to determine the most likely label. LVLMs, such as QWEN2-VL (Wang et al., 2024), contain a Large Language Model (LLM) in themselves and are able to use their vast language knowledge with unified multimodal pre-training to achieve impressive capabilities in zero-shot recognition. However, the accuracy for fine-grained recognition is limited. We aim to improve accuracy by fine-tuning on a subset of categories of a new dataset (called the "base" set) and test on "novel" categories for which no training examples are seen. This setting is common (Zhou et al., 2022b) and relevant for adapting models to new domains with limited training data.

Our approach is based on two key observations. First, the base model exhibits high variance across its `Pass@K` performance (Figure 1a): the correct label often appears among the $K$ sampled response, yet fails to get it correct as `Pass@1`. This indicates possible over-reliance on coarse, salient attributes shared by related categories and may benefit from a fine-grained, differential reasoning to separate semantically similar categories. The second observation is that the LVLMs actually contain detailed knowledge about the parts and attributes of the base and novel categories which could be used for detailed, differential analysis.

Inspired by the success of Chain of Thought (CoT) reasoning (Wei et al., 2022) using Reinforcement Learning (RL) (Guo et al., 2025) for pure language tasks such as mathematics and coding (Shao et al., 2024b; Jiang et al., 2023), ViRFT (Liu et al., 2025) extended this idea to vision tasks such as image classification. The key idea in ViRFT is to construct a verifiable reward (Lambert et al., 2024) for image classification which allows visual reasoning RL training via GRPO (Shao et al., 2024b).

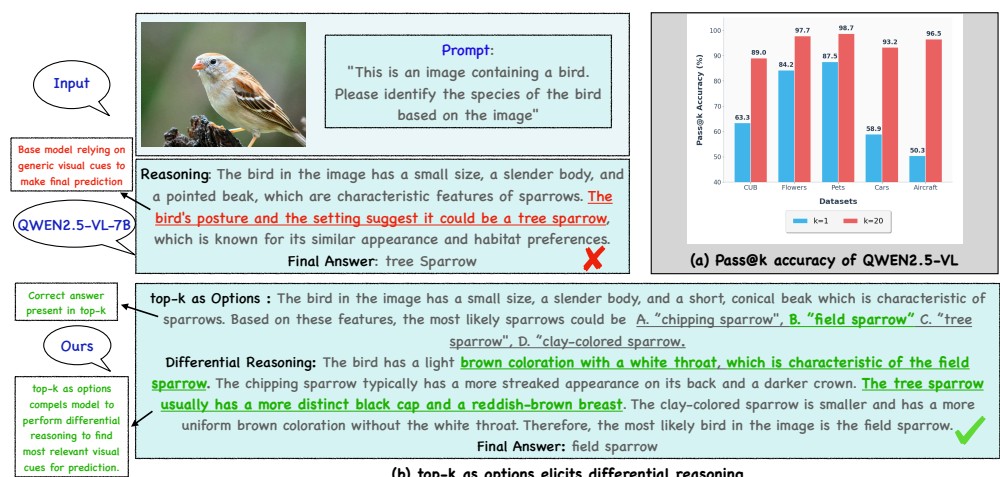

Figure 1: For fine-grained image recognition task, most salient visual attributes are often insufficient to identify the correct category as its common among similar categories. (a) This leads to a significant performance gap in model's `Pass@1` and `Pass@20` accuracy. (b) A differential reasoning can help indicate out the key visual attributes that can help distinguish among similar categories. Base model fails to use such discriminative features relying only on prominent visual features. We solve this by using top-k as options (the most likely categories base model confuses it for) and utilizes model's text knowledge to resolve this confusion using differential reasoning (highlighted in green).

However, their verifiable reward obtained through exact string match between the category name and the model's final answer is brittle: (i) requiring ad hoc string post-processing and model responses may use scientific or common names that cannot be validated by string matching; (ii) encourages memorization of the training category names (Section 4.2.1); and (iii) fails to incentivize attribute-level, discriminative reasoning (Figure 1b). This leads to a weak base to novel generalization and a tendency to ignore useful text knowledge when visual evidences alone are ambiguous.

To overcome these deficiencies, we propose **DiVE-k** framework (**Di**fferential **V**isual r**E**asoning using top-**k** generations) that treats base model's top-k generations, obtained via $K$ rollouts, as training primitive that enables differential visual reasoning. For each training image, we treat the top-k outputs of the base model as an explicit hypotheses set and train the model using RL to resolve this set by selecting the correct element.

We formulate this as a Multiple-Choice-Question (MCQ) interface, using top-k as options leveraging the model's own distribution. This yields two advantages (i) differential reasoning: presenting the model with its own top predictions as options compels it to move beyond simple pattern recognition. Thus, the model learns to engage in fine-grained reasoning, identifying the specific attributes that differentiate the correct answer from other plausible alternatives, as illustrated in Figure 1. (ii) easily verifiable reward signal: the reward for a correct prediction becomes trivially verifiable as model simply has to select the correct index from the given options. This contrasts with methods such as ViRFT, which rely on an exact string match. Our approach further mitigates the category name memorization issue and leads to better generalization on unseen categories.

Experiments on five standard fine-grained image classification datasets show that DiVE-k outperforms existing methods by a significant margin in two distinct zero-shot settings. For standard base-to-novel generalization, our method surpasses pre-trained QWEN2.5-VL-7B and ViRFT by 10.04% and 6.16% on the Harmonic Mean (HM), respectively. This performance gain also extends to mixed-domain zero-shot base-to-novel generalization setting, where we achieve improvements of 9.03% against QWEN2.5-VL-7B and 4.02% against ViRFT. Further, we observe an average improvement of 7.73% compared to ViRFT on 4-shot image classification.

In summary, our main contributions are: (i) we propose DiVE-k framework which uses the top-k generations of base model as a training signal for fine-grained image classification, (ii) we demon-

strate the benefits of using MCQ to distinguish among semantically similar categories, and (iii) we show improved performance on multiple fine-grained image classification datasets with detailed ablation studies.

## 2 PRIOR WORK

**Zero-shot fine-grained image classification** Vision Language Models (VLMs) (Radford et al., 2021; Li et al., 2022; Tschannen et al., 2025; Yuan et al., 2021) use the idea of aligning image with text to achieve zero-shot learning (Lampert et al., 2013; Socher et al., 2013; Wang et al., 2018; Zhang et al., 2017). Although very competent for image-text alignment, these models have limited world knowledge unlike LLMs (Radford et al., 2019; Brown et al., 2020; Touvron et al., 2023; Chowdhery et al., 2023). An early line of research proposes the idea of prompt learning (Zhou et al., 2022a; Khattak et al., 2023b) where a prompt vector is learned for text prompt's context words. Zheng et al. (2024c) uses LLM's knowledge to learn the prompt vector. Another approach to bridge the knowledge gap is by combining the perceptual strengths of VLMs with the linguistic abilities of LLM (Esfandiarpoor & Bach, 2023; Menon & Vondrick, 2022; Pratt et al., 2023; Zeng et al., 2022; Novack et al., 2023; Roth et al., 2023). Our work in part is inspired by FuDD (Esfandiarpoor & Bach, 2023) that uses a multi-stage reasoning by using LLM knowledge to find pairwise discriminative features to complement VLM. However, FuDD generates a fixed set of text prompts offline, limiting its ability to adapt its reasoning strategy to the specific difficulty of each input. Other related work tries to use LLM as a tool for reasoning through programming and language reasoner (Chen et al., 2023; Gupta & Kembhavi, 2023; Zhang et al., 2023; Surís et al., 2023). While these methods improved model performance, the separation between the vision and language modules creates a bottleneck, inhibiting seamless and integrated reasoning across modalities (Liu et al., 2023; 2024).

Recent LVLMs (Bai et al., 2025; Hurst et al., 2024; Team et al., 2023; 2025a) have excellent visual understanding, such as VQA (Shao et al., 2024a), combining visual encoding directly into a LLM architecture. This opens up the possibility of joint vision and text reasoning capability (Team et al., 2025c; Bai et al., 2025; Team et al., 2025a). Additionally, recent success of RL for reasoning on maths and coding tasks (Jaech et al., 2024; Guo et al., 2025; Team et al., 2025b) have transformed the post-training reasoning research and there is a growing interest to extend these ideas to LVLMs.

**Enhancing Reasoning with Reinforcement Learning .** Building upon the seminal work of In-context learning (Brown et al., 2020) and CoT (Wei et al., 2022), the field has moved beyond static prompting by applying RL to fine-tune these reasoning processes (Jaech et al., 2024; Guo et al., 2025; Team et al., 2025b). By treating the generation of a reasoning chain as a sequential decision-making problem, RL-based methods can train models to produce more accurate and explainable solutions. Recent breakthrough in DeepSeek-R1 (Shao et al., 2024b) showed the effectiveness of CoT based training further making it more efficient using their GRPO algorithm. Inspired by these success in text-based reasoning, a growing body of work apply RL to enhance the reasoning capabilities of LVLMs for vision-centric tasks, such as image classification (Liu et al., 2025; Li et al., 2025), Object detection, Grounding (Liu et al., 2025; Shen et al., 2025), and Visual Question Answering (Cao et al., 2025; Sarch et al., 2025; Fan et al., 2025). Within our target domain of fine-grained image recognition, the most pertinent work is ViRFT (Liu et al., 2025), which trains an LVLM using exact string matching reward to foster visual reasoning. However, this reward mechanism proves brittle, failing to generalize in base-to-novel settings. Furthermore, it does not fully leverage the LLM's inherent knowledge about fine-grained categories to incentivize the differential reasoning necessary for distinguishing among similar options. Zhu et al. (2025); Chen et al. (2025) proposes to enhance LLM performance by improving `Pass@k` accuracy. In contrast, we use model's own knowledge of `Pass@k` to improve their reasoning, specifically for vision-language task.

## 3 DIVE-K FRAMEWORK

DiVE-k framework employs a simple two-step strategy which elicits a differential reasoning in LVLM using its top-k generations as training signal. An overview of our proposed method is shown in Figure 2. In the first step (red box in 2), we perform an offline top-k generation using the base model to construct a potential hypotheses set to be used for constructing Multiple Choice Questions (MCQs). In the second step (green box in 2), we use the MCQ dataset for RL training using GRPO.

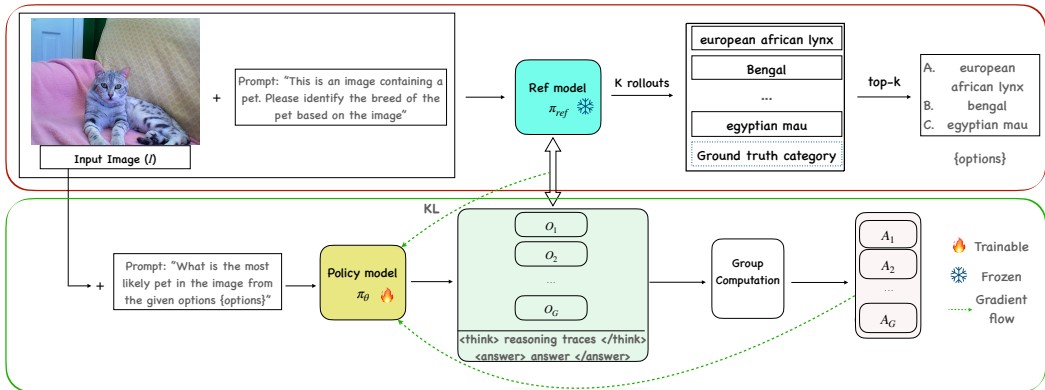

Figure 2: An overview of DiVE-k framework. First, we do an offline option mining (red box) where for each training image, we sample $K$ rollouts from a pretrained LVLM and select top-k options by frequency, ensuring the ground-truth appears. Next we perform RL training using GRPO on MCQ prompts (green box): the model receives an image, a natural language prompt, and k options as input and produces a reasoning chain and a final choice and is optimized with a simple, verifiable reward that combines MCQ correctness and format compliance.

**Top-k as hypotheses set.** In the first step of the DiVE-k framework, given an image $I$ and a text query $q$, we use the base policy model $\pi_\theta$ to rollout $K$ responses $\mathcal{Y} = (y_1, y_2, ..., y_K)$ sampled through a specific decoding strategy (e.g. $top-p$ nucleus sampling).

$$\mathcal{Y} \sim \pi_\theta(I, q, K) \tag{1}$$

Each of these responses ($y_i \in \mathcal{Y}$) can be represented as $y_i = (r_i, c_i)$, where $r_i$ is the reasoning trace, and $c_i$ is the final predicted category name. Let $\mathcal{C}$ be the unique category names set within the $K$ generated responses. Next, we count the frequencies of each category in $\mathcal{C}$ and using this frequency count, we construct the option set $\mathcal{O}_{top-k}$ by selecting the $k$ most frequent categories. The value of $k$ is set to $k = \min(m, |\mathcal{C}|)$, where $|\mathcal{C}|$ is the number of unique categories, and we use $m = 5$. Let $\hat{c}$ be the ground-truth category name. To ensure that the correct answer is always an option during training, we adjust the set if necessary. If $\hat{c} \notin \mathcal{O}_{top-k}$, we modify $\mathcal{O}_{top-k}$ by replacing the least frequent candidate with the ground-truth $\hat{c}$.

Finally, the option set $\mathcal{O}_{top-k}$ is structured into a standard MCQ format. The options are enumerated and assigned labels (e.g., A, B, C, ...). The options are randomly shuffled to avoid any option-order bias. The ground-truth label, $\hat{a}$, is the label corresponding to the correct category $\hat{c}$. Thus, each sample in our final dataset, $\mathcal{D}$, is a tuple $(I, q, \mathcal{O}_{enum}, \hat{a})$, where $I$ is the image, $q$ is the query, $\mathcal{O}_{enum}$ is the enumerated list of option strings, and $\hat{a}$ is the ground-truth label for the correct option.

To focus on challenging examples through hard-negative mining, we filter out trivial cases from the training set. Specifically, we exclude any sample for which the model generates only a single, correct category prediction (i.e., $|\mathcal{C}| = 1$ and $\mathcal{C} = \{\hat{c}\}$). Note that our first step to generate options with the same base model used as the policy model during training is crucial, as it ensures the categories are drawn from the model's own distribution and yields optimal learning as shown in section 4.3.1.

**RL training using GRPO.** We train the model using the MCQ dataset $\mathcal{D}$ constructed in step one. Our task is defined by $\mathcal{D}$ consisting of $(I, q, \mathcal{O}_{enum}, \hat{a})$ as explained in previous section. Our goal is to train LVLM as policy model $\pi_\theta$ which can generate $(s, a)$ where $s$ is the intermediate reasoning tokens and $a$ is final answer. To achieve this, we train the model using GRPO algorithm (Shao et al., 2024b). During training, for every data sample $d_i$, model generate $N$ rollouts $(O_0, O_1, ..., O_N)$ using the current policy model $\pi_\theta$. For each of these responses, reward $(r_0, r_1, ..., r_N)$ are computed. These rewards for each group is then used for group advantage estimation using equation 2

$$A_i = \frac{r_i - \text{mean}\{r_1, \ldots, r_N\}}{\text{std}\{r_1, \ldots, r_N\} + \delta}, \tag{2}$$

where $\delta$ is a small valued constant.

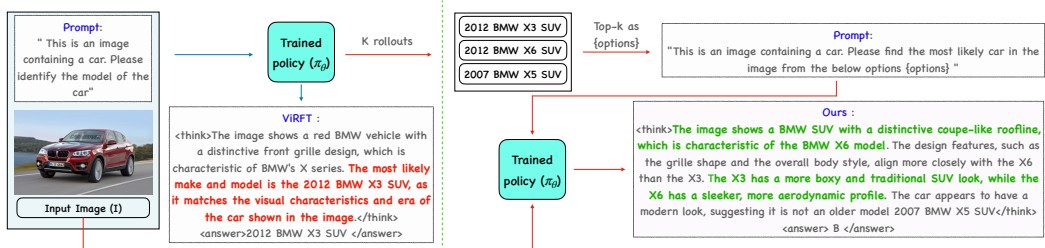

Figure 3: An example to illustrate our inference pipeline (red arrows) and its comparison to existing method (blue arrows). Similar to training phase, we perform inference in two steps (right of dotted line), where we first generate option by choosing top-k responses from $K$ rollouts and then model picks the correct answer among the options unlike open-ended one step inference of existing methods (left of dotted line)

Our reward consists of two parts: first is MCQ reward ($r_{mcq}$) and second is format reward ($r_{format}$). $r_{mcq}$ checks for correctness in the response and rewards a value of 1.0 if model predicts the correct option as answer and 0.0 otherwise as in equation 3

$$r_{\text{mcq}}(\hat{a}, a) = \begin{cases} 1.0, & \text{if } \hat{a} = a \\ 0.0 & \text{otherwise.} \end{cases} \tag{3}$$

$r_{format}$ encourages the correct formatting for the output response to make it easier to extract <think> and <answer> tags. Our final reward ($r$) is defined as the weighted sum of these two: $r = \lambda_f r_{format} + \lambda_m r_{mcq}$. Our training objective function $\mathcal{J}_{\text{GRPO}}(\theta)$ is defined as:

$$\mathcal{J}_{\text{GRPO}}(\theta) = \frac{1}{N} \sum_{i=1}^{N} \left[ \min(s_i A_i, \text{clip}(s_i, 1 - \epsilon, 1 + \epsilon) A_i) - \beta D_{\text{KL}}(\pi_\theta \| \pi_{\text{ref}}) \right] \tag{4}$$

where $\epsilon$ and $\beta$ are the hyperparameters. $\pi_{\text{ref}}$ is the reference policy model (usually the pre-trained model) used to control the divergence of trained policy. $s_i = \frac{\pi_\theta(o_i|q)}{\pi_{\theta_{\text{old}}}(o_i|q)}$, where $\pi_{\theta_{\text{old}}}$ is the old policy model before the update. Overall, this objective function aims to maximize the expected reward while keeping the policy close to the reference policy for stable learning.

**Inference.** Figure 3 shows our inference pipeline and compares it to ViRFT which performs a single-pass inference (left of the dotted line) to directly predict the category names, while we use a two-step pipeline (right of the dotted line) similar to training phase. We use the trained policy model to first generate potential options using top-k generations and then re-prompt the same policy to select the correct option among the provided options. It should be noted that we do not add ground-truth to the options if the first step fails to generate the groundtruth category as an option.

## 4 EXPERIMENTS

### 4.1 EXPERIMENTAL SETUP

**Baselines and Datasets.** We compare our results against various baselines: 1) pre-trained QWEN2.5-VL-7B (Bai et al., 2025) 2) Supervised Fine Tuning (SFT) (Zheng et al., 2024a) using full model and LoRA (Hu et al., 2022) training. 3) ViRFT (Liu et al., 2025) trained under the same base-novel setting. 4) QWEN2.5-VL-7B and ViRFT using our inference pipeline 5) Consistency as accuracy: During the two-step inference process, use the most consistent prediction from $K$ generation as final prediction. We also report the performance of the proprietary models, Gemini2.5-flash-lite(Comanici et al., 2025), GPT-5-mini (OpenAI, 2025) and Grok4-fast (xAI, 2025), with closed weights accessible through APIs (OpenRouter, 2024).

We evaluate our method on five standard fine-grained image classification dataset across various domains, including OxfordFlowers-102 (Nilsback & Zisserman, 2008), CUB-200 (Wah et al., 2011),

OxfordPets-37 (Parkhi et al., 2012), StanfordCars-196 (Krause et al., 2013) and FGVC Aircraft-100 (Maji et al., 2013). For the base-novel split, we follow previous work (Khattak et al., 2023a) to divide the categories into equal halves for base and novel. For instance in CUB-200, the first 100 categories are considered base and the other 100 are considered novel and similarly for other datasets. Note that neither images nor the category names of the novel classes are seen during training.

Table 1: Quantitative comparison with existing methods on zero-shot base-to-novel generalization. Our proposed methods shows strong generalization outperforming existing methods. B → Base, N → Novel , H → Harmonic Mean, Gemini2.5-f-l → Gemini2.5-flash-lite, QWEN2.5 → QWEN2.5-VL-7B. Rows with * are results of existing method using our inference pipeline. Consistency → Most consistent prediction in top-k generation as final answer.

| | Flowers | | | CUB | | | Pets | | | Cars | | | Aircraft | | | Avg | | |
|---|---|---|---|---|---|---|---|---|---|---|---|---|---|---|---|---|---|---|
| Method | B | N | H | B | N | H | B | N | H | B | N | H | B | N | H | B | N | H |
| **Proprietary Models** | | | | | | | | | | | | | | | | | | |
| Gemini2.5-f-l | 94.1 | 91.6 | 92.8 | 75.3 | 60.3 | 67.0 | 91.5 | 96.4 | 93.9 | 73.9 | 92.3 | 82.1 | 59.7 | 65.4 | 62.4 | 78.9 | 81.2 | 80.0 |
| GPT-5-mini | 97.4 | 94.4 | 95.9 | 76.3 | 64.7 | 70.0 | 93.1 | 97.5 | 95.2 | 82.0 | 94.7 | 87.9 | 60.2 | 74.5 | 66.6 | 81.8 | 85.1 | 83.4 |
| Grok-4-fast | 81.3 | 87.4 | 84.2 | 63.3 | 50.7 | 56.3 | 84.1 | 94.2 | 88.8 | 72.2 | 86.6 | 78.7 | 44.0 | 55.4 | 49.1 | 69.0 | 74.8 | 71.8 |
| **Method Comparison** | | | | | | | | | | | | | | | | | | |
| CLIP | 72.1 | 77.8 | 74.8 | 65.5 | 48.7 | 55.8 | 91.2 | 97.3 | 94.1 | 63.4 | 74.9 | 68.6 | 27.2 | 36.3 | 31.1 | 63.8 | 67.0 | 64.9 |
| QWEN2.5 | 84.2 | 83.8 | 84.0 | 63.3 | 48.2 | 54.7 | 87.5 | 93.3 | 90.3 | 58.9 | 72.9 | 65.1 | 50.3 | 54.3 | 52.3 | 68.9 | 70.5 | 69.7 |
| SFT (Full) | 93.7 | 62.6 | 75.1 | **84.8** | 29.2 | 43.4 | **95.8** | 42.6 | 59.0 | **81.2** | 40.6 | 54.1 | 64.7 | 10.1 | 17.5 | **84.0** | 37.0 | 49.8 |
| SFT (LoRA) | 84.5 | 83.9 | 84.2 | 62.2 | 49.2 | 54.9 | 85.7 | 93.3 | 89.3 | 59.9 | 71.2 | 65.0 | 49.5 | 54.3 | 51.8 | 68.3 | 70.4 | 69.3 |
| ViRFT | 84.3 | 84.6 | 84.5 | 65.4 | 51.0 | 57.3 | 90.5 | 95.5 | 92.9 | 60.3 | 73.6 | 66.3 | 64.6 | 66.3 | 65.4 | 73.0 | 74.2 | 73.6 |
| QWEN2.5* | 90.7 | 87.9 | 89.3 | 68.5 | 58.7 | 63.2 | 85.6 | 93.8 | 89.5 | 63.3 | 76.6 | 69.4 | 63.3 | 68.0 | 65.6 | 74.3 | 77.0 | 75.6 |
|   Consistency | 85.9 | 85.5 | 85.7 | 64.2 | 54.0 | 58.6 | 90.2 | 94.7 | 92.4 | 62.1 | 72.0 | 66.7 | 61.8 | 63.5 | 62.6 | 72.8 | 74.0 | 73.2 |
| ViRFT* | 89.5 | 87.9 | 88.7 | 70.0 | 60.0 | 64.8 | 92.6 | 92.8 | 92.7 | 64.8 | **76.9** | 70.4 | 66.4 | 67.8 | 67.1 | 76.8 | 77.1 | 76.9 |
|   Consistency | 85.3 | 85.2 | 85.3 | 67.5 | 57.2 | 61.9 | 92.0 | **96.1** | 94.0 | 64.8 | 73.2 | 68.8 | 67.4 | 64.2 | 65.7 | 75.4 | 75.2 | 75.1 |
| DiVE-k (ours) | **97.4** | **88.9** | **92.9** | 80.5 | **65.5** | **72.2** | 89.1 | 94.2 | 91.6 | 69.0 | 76.2 | **72.4** | **68.1** | **69.1** | **68.6** | 80.8 | **78.8** | **79.8** |
|   Consistency | 95.3 | 85.6 | 90.2 | 75.2 | 62.3 | 68.2 | 93.1 | 95.5 | **94.3** | 68.7 | 74.4 | 71.4 | 68.0 | 65.6 | 66.8 | 80.0 | 76.7 | 78.2 |
| Δ vs ViRFT | +13.1 | +4.3 | +8.4 | +15.1 | +14.5 | +14.9 | -1.4 | -1.3 | -1.3 | +8.7 | +2.6 | +6.1 | +3.5 | +2.8 | +3.2 | +7.8 | +4.6 | +6.2 |
| Δ vs QWEN2.5 | +13.2 | +5.1 | +8.9 | +17.2 | +17.3 | +17.5 | +1.6 | +0.9 | +1.3 | +10.1 | +3.3 | +7.3 | +17.8 | +14.8 | +16.3 | +11.9 | +8.3 | +10.1 |

**Evaluation setting and Metric.** We evaluate our method under two distinct zero-shot settings. The first is the standard zero-shot base-to-novel generalization, following (Zheng et al., 2024b), where a separate model is trained on the base classes of each dataset. The second is our proposed mixed-dataset setting, designed to assess cross-domain generalization capabilities. Here, a single model is trained on a unified dataset constructed by combining the base classes from all datasets. Additionally, we also evaluate our method under few-shot classification setting with 4 shots per class. For performance measurement, we report the classification accuracy on base and novel classes, along with their Harmonic Mean (HM).

For evaluation, we use the LLM gemini-2.5-flash-lite (Comanici et al., 2025), where we provide the ground-truth category name and model predicted category name and ask the LLM if they belong to the same fine-grained category or not. This specifically helps us evaluate better for the answers where model responds a scientific name and the provided ground-truth is common name and vice-versa. We provide more details in Appendix A.1.2

**Implementation Details.** We use the pre-trained Qwen2.5-VL-7B-Instruct (Bai et al., 2025) as our base model and perform the RL training using the proposed method. We use K=20 during offline option generation using $K$ rollouts and other details are provided in Appendix A.1.1. For Zero Shot Base-to-Novel training, we train the model for 400 steps, whereas for mixed data training, we train it for 1 epoch. For few-shot training, we train each model for 200 steps following ViRFT. All models are trained on three A6000 GPUs with 48GB memory with an overall batch size of 6. For GRPO, a total of 4 responses are generated for each input sample. Following ViRFT, we use learning rate of $10^{-6}$, AdamW optimizer, linear scheduler and set $\lambda_m = \lambda_f = 1$. For ViRFT training, we use their official code from github with some modification discussed in Appendix A.1.1.

## 4.2 RESULTS

### 4.2.1 ZERO-SHOT BASE-TO-NOVEL GENERALIZATION

We present the results for zero-shot base-to-novel generalization in Table 1. Across five fine-grained benchmarks, DiVE-k shows the best generalization, yielding the highest average harmonic mean (HM) of 79.8 with average base/novel accuracies of 80.8/78.8. Relative to the strongest baseline

(ViRFT), this corresponds to gains of +7.8 base, +4.6 novel, and +6.2 HM. Even when baselines are run with our inference pipeline (ViRFT*), we retain a +4.0 base, +1.8 novel improvements. Notably, under our inference pipeline, ViRFT gain over QWEN2.5-VL-7B model on novel categories remains marginal by +0.1 (77.0→77.1), suggesting their reward primarily reinforces base categories. In contrast, our approach delivers a substantive +1.8 boost on novel classes under the same inference setting, indicating better generalization beyond the training categories.

The improvements are especially pronounced on CUB (+14.9 HM) and Oxford Flowers (+8.5 HM), and remain consistent on Stanford Cars (+6.1 HM) and FGVC Aircraft (+3.2 HM), indicating robust zero-shot transfer. We observe a small regression on Pet data compared to ViRFT, possibly due to more options leading the model to make more mistake during the second step. We show that our method outperforms ViRFT for smaller $K$ and provide more analysis for this in Ablation 4.3.2.

We also evaluate supervised fine-tuning (SFT) as a baseline. Although SFT yields strong accuracy gains on base categories, its performance deteriorates sharply on novel categories, with an average accuracy drop of 33.5% relative to the base model and a 19.9% reduction in HM. This sharp degradation highlights SFT's inability to generalize under the base-to-novel transfer setting. To provide a broader perspective, we also include results from proprietary models which are likely much larger. In this context, our method surpasses Grok4-fast and is on par with Gemini2.5-flash-lite, though GPT-5-mini leads. We provide more results using Gemma-3 (Team et al., 2025a) base model in Appendix A.3.

Table 2: Quantitative comparison of our method with baselines under mixed-dataset base-to-novel generalization. QWEN2.5 → QWEN2.5-VL-7B. Rows with * are results of existing method using our inference pipeline. Consistency → Most consistent prediction in top-k generations as output.

| Method | Flowers | | | CUB | | | Pets | | | Cars | | | Aircraft | | | Avg | | |
|---|---|---|---|---|---|---|---|---|---|---|---|---|---|---|---|---|---|---|
| | B | N | H | B | N | H | B | N | H | B | N | H | B | N | H | B | N | H |
| QWEN2.5 | 84.2 | 83.8 | 84.0 | 63.3 | 48.2 | 54.7 | 87.5 | 93.3 | 90.3 | 58.9 | 72.9 | 65.1 | 50.3 | 54.3 | 52.3 | 68.9 | 70.5 | 69.7 |
| ViRFT | 87.2 | 85.9 | 86.5 | 65.7 | 54.0 | 59.3 | 90.5 | 95.0 | 92.7 | 59.7 | 73.6 | 66.0 | **67.2** | 68.0 | **67.5** | 74.1 | 75.3 | 74.7 |
| QWEN2.5* | 90.7 | 87.9 | 89.3 | 68.5 | 58.7 | 63.2 | 85.6 | 93.8 | 89.5 | 63.3 | 76.6 | 69.4 | 63.3 | 68.0 | 65.6 | 74.3 | 77.0 | 75.6 |
| Consistency | 85.9 | 85.5 | 85.7 | 64.2 | 54.0 | 58.6 | 90.2 | 94.7 | 92.4 | 62.1 | 72.0 | 66.7 | 61.8 | 63.5 | 62.6 | 61.9 | 67.8 | 64.6 |
| ViRFT* | 90.2 | 87.9 | 89.0 | 71.3 | 58.3 | 64.2 | 90.2 | 91.6 | 90.9 | 63.7 | 76.6 | 69.6 | 66.9 | 66.4 | 66.7 | 76.5 | 76.2 | 76.1 |
| Consistency | 85.2 | 85.1 | 85.2 | 68.0 | 57.5 | 62.3 | 91.5 | **96.1** | 93.7 | 65.6 | 74.2 | 69.7 | 63.2 | 65.5 | 64.3 | 64.4 | 69.9 | 67.0 |
| DiVE-k (ours) | **97.4** | **89.9** | **93.5** | **76.8** | **61.3** | **68.2** | 87.8 | 94.7 | 91.1 | **68.5** | **78.5** | **73.1** | 65.5 | **69.7** | 67.5 | **79.2** | **78.8** | **78.7** |
| Consistency | 92.8 | 86.0 | 89.3 | 67.7 | 56.7 | 61.7 | **91.8** | **96.1** | **93.9** | 59.6 | 72.5 | 65.4 | 61.1 | 63.5 | 62.3 | 60.4 | 68.0 | 63.9 |
| Δ vs ViRFT | +10.2 | +4.0 | +7.0 | +11.1 | +7.3 | +8.9 | -2.7 | -0.3 | -1.6 | +8.8 | +4.9 | +7.1 | -1.7 | +1.7 | 0.0 | +5.1 | +3.5 | +4.0 |
| Δ vs QWEN2.5 | +13.2 | +6.1 | +9.5 | +13.5 | +13.1 | +13.5 | +0.3 | +1.4 | +0.8 | +9.6 | +5.6 | +8.0 | +15.2 | +15.4 | +15.2 | +10.3 | +8.3 | +9.0 |

### 4.2.2 MIXED-DATASET BASE-TO-NOVEL GENERALIZATION

Training a single model on the union of base categories from all five datasets provide a strong evaluation of mixed-dataset generalization. Table 2 shows the quantitative results for this setting. Our method attains the highest average harmonic mean (HM) of 78.7, improving over the pretrained QWEN2.5-VL-7B by +9.0 HM and over ViRFT by +4.0 HM, with average base/novel gains of +5.1/+3.5. Under our two-step inference on novel classes, ViRFT underperforms the pretrained QWEN2.5-VL-7B (76.2 vs. 77.0), while our approach reaches 78.8, a +1.8 improvement. This contrast suggests that ViRFT struggles to transfer when trained on the mixed base corpus, likely reinforcing base-only cues, whereas our method maintains robust generalization to unseen categories.

Table 3: Quantitative comparison of our proposed method under 4-shot setting.

| Model | Oxford Flowers | CUB | Oxford Pets | Stanford Cars | FGVC Aircraft | Average |
|---|---|---|---|---|---|---|
| QWEN2.5-VL-7B | 78.43 | 51.62 | 79.05 | 57.91 | 52.50 | 63.90 |
| ViRFT | 81.12 | 51.75 | 85.81 | 57.65 | 58.75 | 67.02 |
| QWEN2.5-VL-7B* | 85.04 | 56.87 | 80.40 | 63.46 | 64.67 | 70.09 |
| consistency | 81.86 | 53.00 | 85.81 | 58.30 | 56.36 | 67.07 |
| ViRFT* | 84.80 | 60.00 | 83.10 | 64.75 | 65.44 | 71.62 |
| consistency | 83.33 | 54.12 | **87.83** | 59.02 | 54.71 | 67.80 |
| DiVE-k (Ours) | **88.72** | **63.87** | 85.14 | **66.90** | **69.10** | **74.75** |
| consistency | 85.78 | 58.00 | 86.48 | 60.17 | 64.92 | 71.07 |
| Δ w.r.t ViRFT | 7.60 | 12.12 | -0.67 | 9.25 | 10.35 | 7.73 |
| Δ w.r.t QWEN2.5-VL-7B | 10.29 | 12.25 | 6.09 | 8.99 | 16.60 | 10.85 |

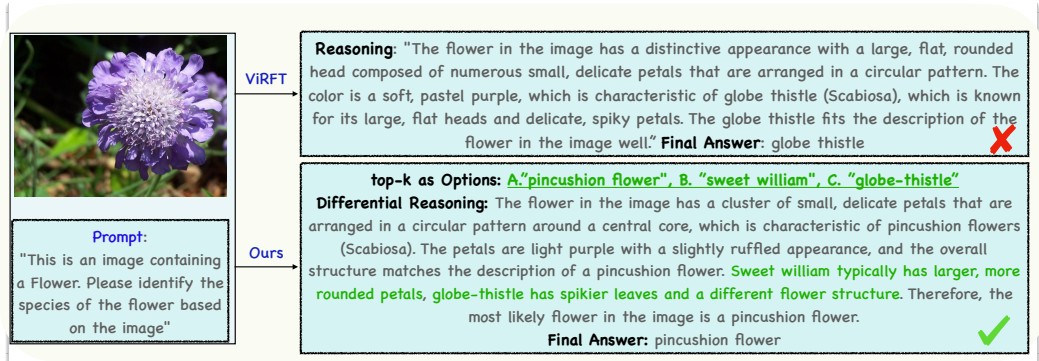

Figure 4: Qualitative comparison on fine-grained flower recognition (top: ViRFT; bottom: Ours). Top: ViRFT predicts "globe thistle," which is incorrect and reflects a coarse judgment. Bottom: Our method enumerates close candidates and uses attribute-grounded, differential reasoning such as capitulum/head shape, floret density and arrangement, bract patterning to select the correct fine-grained label with a justification aligned to the final choice.

### 4.2.3 FEW-SHOT CLASSIFICATION

DiVE-k also shows improvement under few-shot classification setting across datasets. As shown in Table 3, our method achieves an average HM classification accuracy of 74.75%, an improvement of 7.73% compared to ViRFT and 10.85% compared to QWEN2.5-VL-7B model, demonstrating the effectiveness of our method even under data efficient training.

### 4.2.4 VISUALIZATION

In Figure 4, we visualize and compare DiVE-k to ViRFT and defer additional visualizations to Appendix A.2. We find that ViRFT directly latches onto high-level "thistle-like" cues and commits to an incorrect category ("global thistle"), without checking the discriminative attributes that separate near-neighbors. In contrast, DiVE-k first proposes a small top-k shortlist and then rules candidates in/out through explicit, attribute-level comparisons, such as head geometry, the density/arrangement of florets, and the presence/shape of bracts, before committing to a final option. This process yields the correct species-level label together with a rationale that stays consistent with the answer reducing overgeneralization and improving interpretability.

### 4.3 ABLATION STUDIES

#### 4.3.1 THE EFFICACY OF top-k FOR MCQ OPTION GENERATION.

Our ablation into the MCQ option generation reveals that the option construction strategy is critical for model's performance. As detailed in Table 4, randomly selecting categories proved suboptimal, yielding only marginal gains and failing to instill robust reasoning capabilities. While employing a text embedding model from gemini embeddings (Lee et al., 2025) to generate semantically similar options offered some improvement, our proposed top-k as options proves significantly more effective. By sampling options directly from the same base model's top-k generations achieves a substantial classification accuracy gain of 12% on base classes and 6.9% on novel classes compared to base model. This demonstrates that leveraging the base model's own knowledge distribution to generate options is the optimal strategy for training, leading to superior generalization on both base and novel sets.

Table 4: Quantitative comparison of classification accuracy on CUB dataset when options are sampled in different ways.

|  | Base | Novel | HM |
|---|---|---|---|
| QWEN2.5-VL-7B | 68.5 | 58.7 | 63.2 |
| Random MCQ | 72.5 | 57.5 | 64.1 |
| Text Emb MCQ | 76.0 | 59.3 | 66.8 |
| `top-k` MCQ | 80.5 | 65.5 | 72.2 |

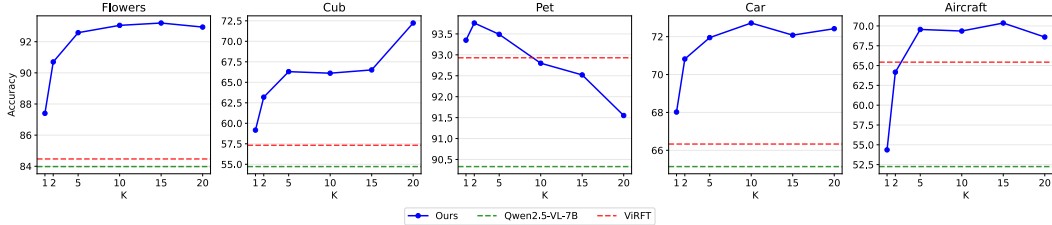

Figure 5: Change in classification accuracy for different values of $K$ on different dataset. Our approach consistently outperforms the baselines for nearly all $K$.

### 4.3.2 ROLE OF TOP-K GENERATIONS DURING INFERENCE

We analyze the impact of the hyperparameter $K$, during $K$ rollout for top-k generation, on classification accuracy, with results presented in Figure 5. The plots show the HM of accuracy across base and novel split for different values of $K$ during inference. Our method consistently surpasses both the baselines at nearly all $K$, often by a large margin. As $K$ increases, accuracy generally rises and then saturates around $K \sim 10 - 15$, yielding near-maximal performance at $K = 15 - 20$ for Flowers ($\sim$93%), CUB ($\sim$72.5%), Car ($\sim$73%), and Aircraft ($\sim$70%), indicating diminishing returns beyond $K = 10$–15. However, for Pet dataset it peaks at small $K$ (93.8% at $K = 2$) and gradually declines for larger $K$, suggesting more options during MCQ leads to more mistakes for this specific dataset. Overall, these results show that our approach achieves improved performance even for small $K$ and thus can get benefit while avoiding extra computation.

### 4.3.3 ROLES OF VISION AND TEXT COMPONENTS

To investigate the individual contributions of the model's vision and text components to its reasoning capabilities, we conduct an ablation to determine whether performance gains stem primarily from updating the vision features, refining the language model's ability to generate a correct chain of reasoning tokens, or a combination of both. The results of which are presented in Table 5.

Table 5: Ablation on effect of training different part of the model on classification accuracy for CUB dataset; Full model training yields the best performance.

|                      | Base  | Novel | HM    |
|----------------------|-------|-------|-------|
| QWEN2.5-VL-7B        | 68.50 | 58.67 | 63.21 |
| Vision only training | 74.00 | 57.83 | 64.92 |
| Text only training   | 74.33 | 60.33 | 66.60 |
| Full model training  | 80.50 | 65.50 | 72.23 |

Our findings reveal distinct roles for each modality. When we fine-tuned only the vision components (freeze text decoder), the model's performance improved significantly on the base dataset (68.5→74) but failed to generalize to the novel dataset, where performance slightly degraded (58.67→57.83). This suggests that adapting visual features alone is insufficient for robust reasoning on new, unseen data. Conversely, training only the text components (freeze vision tower) improved performance on both the base (68.50→74.33) and novel (58.67→60.33) sets. This indicates that enhancing the language model's ability to generate logical reasoning is critical for generalization. However, the best performance was achieved through full model training, which yielded substantial gains on both base (68.50→80.50) and novel (58.67→65.50) sets. This demonstrates that while language model adaptation is key, a combination of both vision and text modules is necessary to unlock the model's full reasoning capability.

### 4.3.4 ACCURACY ANALYSIS OF THE TWO STEPS IN DiVE-K

As DiVE-k framework operates in two stages, the second stage can only succeed if the correct answer appears in the top-k rollouts from the first stage. Therefore, understanding the bottleneck between these two steps is crucial for diagnosing model performance.

Table 6 reports the top-k accuracy (Step 1) of both the base QWEN2.5-VL-7B and the DiVE-k. We find that the base model already achieves strong top-k recall across most datasets, indicating that the correct answer is typically recoverable via sampling. DiVE-k training further improves this

recall, yielding more consistent retrieval of the correct candidates. This improvement is particularly beneficial because Step 2 can only operate correctly when Step 1 supplies the correct option.

Table 6: Top-k accuracy (step 1) across five datasets

| Method | Flowers | | CUB | | Pets | | Cars | | Aircraft | | Avg | |
|---|---|---|---|---|---|---|---|---|---|---|---|---|
| | B | N | B | N | B | N | B | N | B | N | B | N |
| QWEN2.5-VL | 96.84 | 92.62 | 83.33 | 74.66 | 98.40 | 98.89 | 86.67 | 90.84 | 89.02 | 90.15 | 90.9 | 89.4 |
| DiVE-k (ours) | 98.99 | 92.62 | 89.66 | 77.50 | 98.94 | 99.16 | 89.74 | 90.57 | 92.96 | 90.21 | 94.1 | 90.0 |
| Δ | 2.15 | 0.00 | 6.33 | 2.84 | 0.54 | 0.27 | 3.07 | -0.27 | 3.94 | 0.06 | 3.2 | 0.6 |

Table 7 presents the Step 2 MCQ accuracy (differential reasoning). We observe an average improvement of 4.3% on base and 1.6% on novel categories. Since Step 2 operates over the candidates generated in Step 1, high top-k recall directly strengthens the effectiveness of differential reasoning. Improvements in candidate quality and reasoning accuracy reinforce each other, resulting in a compounding effect on the final classification performance.

Across most datasets, Step 1 accuracy is already high (often above 90%), which shifts the primary performance bottleneck to the differential reasoning stage. DiVE-k explicitly targets this challenge and achieves clear gains in MCQ accuracy. For the CUB dataset, however, top-k accuracy remains comparatively lower, leaving additional room for improvement in Step 1. This highlights that the two stages contribute differently depending on dataset difficulty. Some benchmarks are limited by candidate generation, while others are limited by reasoning over those candidates.

Table 7: MCQ accuracy (Step 2) across five datasets

| Method | Flowers | | CUB | | Pets | | Cars | | Aircraft | | Avg | |
|---|---|---|---|---|---|---|---|---|---|---|---|---|
| | B | N | B | N | B | N | B | N | B | N | B | N |
| QWEN2.5-VL | 93.65 | 94.90 | 82.20 | 78.62 | 86.99 | 94.85 | 73.03 | 84.32 | 71.10 | 75.42 | 81.4 | 85.6 |
| DiVE-k (ours) | 98.40 | 95.98 | 89.78 | 84.52 | 90.05 | 95.00 | 76.90 | 84.10 | 73.25 | 76.59 | 85.7 | 87.2 |
| Δ | 4.75 | 1.08 | 7.58 | 5.90 | 3.06 | 0.15 | 3.87 | -0.22 | 2.15 | 1.17 | 4.3 | 1.6 |

## 5 LIMITATIONS AND FUTURE WORK

While DiVE-k framework effectively leverages the model's intrinsic knowledge acquired during pre-training to improve its downstream accuracy, our two-step inference process incurs additional computational cost due to the requirement of two forward passes. The success of our approach is demonstrated on QWEN2.5-VL, which exhibits high initial `Pass@k` accuracy. However, the method's efficacy is contingent on this baseline performance; base LVLMs with lower intrinsic accuracy may not realize comparable gains. A potential direction to mitigate this dependency, which we leave for future work, is to incorporate a `Pass@k` accuracy as reward signal directly into the training objective. Another promising avenue for future research involves the verification of generated reasoning traces for factual correctness and their grounding in the input image.

## 6 CONCLUSION

In conclusion, we introduced DiVE-k, a novel framework that addresses the limitations of Large Vision Language Models in fine-grained image recognition. By utilizing top-k generations as training primitive, our method requires the model to perform differential reasoning among visually similar categories using a multiple-choice question format. Extensive experiments across five standard datasets demonstrate that DiVE-k significantly outperforms existing approaches in base-to-novel generalization, mixed domain, and few-shot settings. Our ablation studies further reveal that the efficacy of this approach hinges on mining options from the base model's own distribution, which is critical for effective RL training. Moreover, we show that the joint fine-tuning of both vision and text components is essential for unlocking the model's full reasoning potential and that increasing the value of k offers diminishing return during inference. Overall, our work highlights the effectiveness of leveraging a model's inherent knowledge distribution to refine its reasoning capabilities, establishing a new direction for improving visual discrimination in LVLMs.

## 7 REPRODUCIBILITY STATEMENT

We aim to ensure full reproducibility of our work by providing detailed descriptions of our methodology, training and inference pipeline, and evaluation protocol in the method, results and appendix section. All hyperparameters, implementation details, and generation settings (e.g., temperature, sampling strategies, and reward design) are listed in the Appendix, along with the exact prompt templates used for all experiments. Experiments are conducted on both open-source and closed-source models; for open-source models, we provide precise checkpoint versions. We plan to publicly release our codebase, experimental configurations, and trained model checkpoints upon acceptance to support reproducibility.

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

# A   APPENDIX

## A.1   IMPLEMENTATION DETAILS

### A.1.1   TRAINING DETAILS

**Sampling parameters.** In the first step of our pipeline, we generate $K$ responses using top-p nucleus sampling. We provide the details about the parameters used during this sampling in Table 8.

Table 8: Generation Arguments

| Parameter | Value |
|---|---|
| `max_new_tokens` | `1024` |
| `temperature` | `1.0` |
| `top_p` | `0.95` |
| `do_sample` | `True` |
| `num_return_sequences` | `20` |
| `repetition_penalty` | `1.1` |

**ViRFT model training.** During our evaluation of ViRFT method, we find a crucial issue in their implementation. The string match logic used during both training (for reward computation) and evaluation (for answer correctness) is shown in Listing A.1.1. We find that the second part of the or statement (student_answer in ground_truth) leads to a shortcut specifically for fine-grained image classification. For example, even if the model responds "gull" this reward function (and evaluation) will consider it as correct even though it doesn't give a correct answer. (it could be any of "california gull", "Heermann Gull", "ivory gull" etc.). This leads to a huge drop in accuracy when evaluated using our LLM evaluation as demonstrated in Table 9 for CUB and Stanford Cars dataset. We fix this issue by removing this shortcut and keeping only the "ground_truth in student_answer" during training for reward computation. After fixing this error we observe expected results shown in 9.

Table 9: Table to demonstrate the shortcut issue in original ViRFT code. ViRFT refers to the accuracy using the original code, ViRFT! refers to our modified code used for training

| Method | CUB | | Stanford cars | |
|---|---|---|---|---|
| | **Base** | **Novel** | **Base** | **Novel** |
| QWEN2.5-VL-7B | 63.33 | 48.17 | 58.87 | 72.9 |
| ViRFT | 39.33 | 27.83 | 13.79 | 23.09 |
| ViRFT! | 65.44 | 51.00 | 60.34 | 73.63 |

**String matching code in ViRFT**

```
# reward computation code
if ground_truth in student_answer or student_answer in ground_truth:
    reward = 1.0

# evaluation code
if image_cate in answer_content or answer_content in image_cate:
    right_count += 1
else:
    print('no')
```

### A.1.2   EVALUATION DETAILS

Here we provide the details of the different prompt used at different stage of our method. In Figure 7 we have provided the complete prompt used in first step of our proposed method. Here,

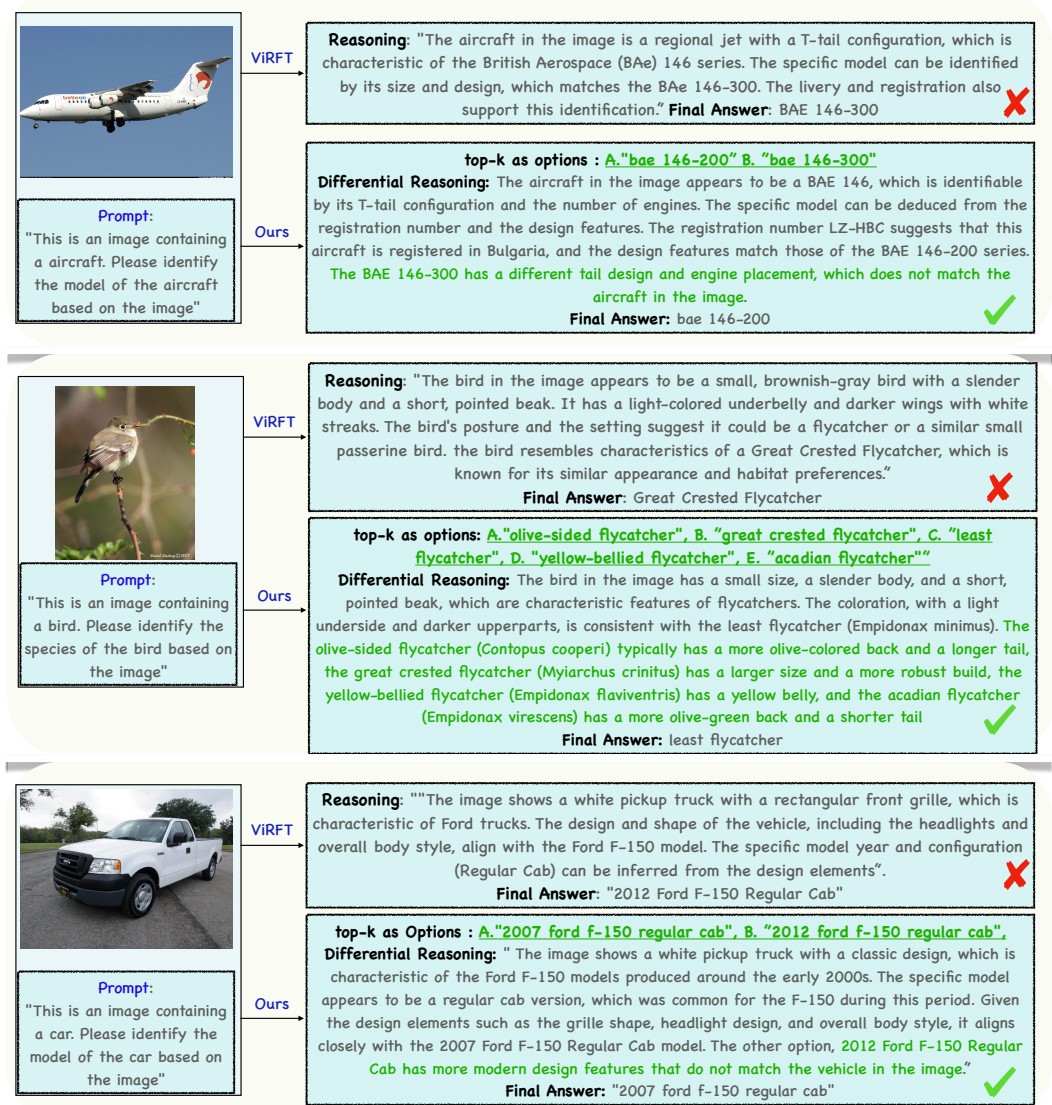

Figure 6: Qualitative comparison across three domains—aircraft, birds, and cars. In each pair, ViRFT commits to a plausible but incorrect class with generic rationale such as "BAe 146-300", "Great Crested Flycatcher", "2012 Ford F-150". Our method first enumerates top-k candidates and then applies attribute-grounded differential reasoning such as T-tail/registration cues for BAe 146-200; Empidonax traits for Least Flycatcher; grille/headlight era cues for a 2007 F-150, yielding the correct fine-grained label and a justification aligned with the final choice.

{category_list} refers to the list of all the category list for the given dataset. During training, we only use base categories in the prompt.

In Figure 8, we provide the prompt used for second step of multiple choice question. Here {options} refers to the options obtained from the first step of our pipeline. In Figure 9, we provide the prompt used during the evaluation. {groundtruth} refers to the groundtruth category name and {prediction} refers to the model's predicted answer. We use `google/gemini-2.5-flash-lite-preview-06-17` as LLM for evaluation from openrouter (OpenRouter, 2024) API.

<image> This is an image containing a bird. Output the most likely species name in the image. The species name of the bird strictly belongs to below category list {category_list}.

Output the thinking process in <think> </think> and final answer in <answer> </answer> tags.

The output answer format should be as follows: <think> ... </think> <answer> species name </answer>.

Please strictly follow the format.

Figure 7: Prompt for first step of our pipeline to generate $K$ rollouts

This is an image containing a bird. Please find the most likely bird in the image from the below options. {options}.

Please output the letter corresponding to the correct category name. Output the thinking process in <think> </think> and final answer in <answer> </answer> tags.

The output answer format should be as follows: <think> ... </think> <answer>option letter</answer>

Please strictly follow the format.

Figure 8: Prompt for Multiple Choice Question (MCQ) answering

## A.2 QUALITATIVE COMPARISON

We provide additional qualitative comparisons in Figure 6. We note that explicit candidate enumeration followed by differential, attribute-level reasoning improves fine-grained recognition. In each case, the top rows (ViRFT) select look-alike but wrong categories—over-generalizing to BAe 146-300, misidentifying a flycatcher species, and over-estimating the truck's model year—supported by broad, non-discriminative explanations. The bottom rows (Ours) surface a short top-k list and then contrast salient cues (e.g., tail/engine/registration details for BAe 146-200; size/underparts/Empidonax patterns for Least Flycatcher; grille and headlight silhouette for a 2007 F-150) before committing to a final answer. This two-step structure reduces over-generalization and aligns the selected label with evidence visible in the image.

Table 10: Comparison of classification accuracy using Gemma3-12B base model on CUB and Flowers datasets.

| Method | CUB | | | Flowers | | |
|---|---|---|---|---|---|---|
| | Base | Novel | HM | Base | Novel | HM |
| Gemma3-12B | 38.67 | 32.00 | 35.03 | 71.26 | 86.52 | 78.16 |
| Gemma3-12B* | 46.00 | 37.17 | 41.12 | 69.54 | 86.67 | 77.17 |
| DiVE-k (Ours) | 58.50 | 40.83 | 48.10 | 84.63 | 87.38 | 85.99 |
| $\Delta$ w.r.t Gemma3 | 20.17 | 8.83 | 13.07 | 13.37 | 0.86 | 7.83 |

## A.3 ADDITIONAL BACKBONE RESULTS

To assess the model-agnostic nature of our approach, we also report results with replacing the QWEN2.5-VL backbone with Gemma3-12B and repeat the evaluation on CUB and Flowers. As shown in Table 10, DiVE-k consistently yields substantial improvements over the Gemma3-12B baseline across all splits. Overall, DiVE-k consistently improves the harmonic mean (HM) performance on both datasets, achieving gains of +13.07 and +7.83 points over the base Gemma3-12B model. These results demonstrate that our framework generalizes beyond a specific foundation model and delivers consistent performance gains across different architectures and datasets.

You are evaluating fine-grained image classification results.
Given:

- Groundtruth category: {groundtruth}
- LLM prediction: {prediction}
Check if the groundtruth matches the prediction. The strings need not match exactly but

they must refer to the same specific fine-grained category, not just broad class.
Respond with:

1. "True" or "False" if groundtruth matches the prediction in `<answer></answer>` tag.
i.e `<answer>`answer here (True/False)`</answer>`
2. Brief explanation in `<explanation></explanation>` tag. i.e
`<explanation>`Explanation here`</explanation>`

Figure 9: Prompt for evaluating fine-grained image classification results.

## A.4 COMPUTATION COST COMPARISON

Since DiVE-k uses a two-step inference pipeline, it incurs additional computational overhead due to two forward passes. To quantify this overhead, we measure the per-sample inference time averaged over 500 samples on A6000 GPUs (48GB). As shown in Table 11, the average per-sample inference time increases from 2.50s in the one-step setting to 12.95s in the two-step setting. However, these improvements are not merely a byproduct of increased computation. Even under an identical compute budget, using K=1, which reduces our method

Table 11: Per-sample inference time comparison between one-step and two-step pipelines.

|  | One Step | Two Steps |
|---|---|---|
| Flowers | 2.50 s | 16.02 s |
| CUB | 2.77 s | 14.58 s |
| Pet | 2.23 s | 8.25 s |
| Average | 2.50 s | 12.95 s |

to a single forward pass with greedy decoding, directly comparable to prior one-step baselines, DiVE-k still outperforms existing methods on 4 out of 5 datasets (Figure 5). This demonstrates that the gains arise from our formulation itself rather than additional compute alone. When more computation is permitted, performance further scales with K, providing a controllable trade-off between inference cost and accuracy that can be adapted to different application constraints.

## A.5 LARGE LANGUAGE MODELS (LLMs) USAGE DETAIL.

We utilized LLMs as a writing aid. Their application was strictly limited to proofreading for errors and polishing the prose for clarity and style. LLMs were not used for any substantive tasks, including but not limited to research, information retrieval, discovery, or the ideation of concepts and conclusions presented herein. All intellectual content is the original work of the authors.

