# OpenReview forum: "DiVE-k: DIFFERENTIAL VISUAL REASONING FOR FINE-GRAINED IMAGE RECOGNITION"
_ICLR.cc/2026/Conference — ICLR 2026 Poster_

### Official Review · Reviewer_CNJv · 2025-10-20

**Soundness:** 3
**Presentation:** 3
**Contribution:** 2
**Rating:** 6
**Confidence:** 3

**Summary:**

The paper introduces the DiVE-k framework, which significantly enhances the base model's fine-grained image recognition capabilities by leveraging its top-k prediction results to construct Multiple-Choice Question (MCQ) data, and subsequently fine-tuning the model using the GRPO algorithm for Reinforcement Learning.

**Strengths:**

1. Using the model's own top-k predictions as training data is interesting and insightful, which serves as a form of hard-negative mining against the model's confusions.
2. DiVE-k achieves significant performance improvements over baseline models across multiple datasets and tasks.
3. The overall writing and presentation of the paper is good.

**Weaknesses:**

1. Limited comparison with related work: The paper primarily contrasts its results with ViRFT. To comprehensively validate the efficacy of the proposed method, it should be compared against additional approaches mentioned in the "Related Work" section.
2. Lack of diverse backbone models: The current experiments are exclusively conducted on Qwen2.5-VL. It is crucial to demonstrate the generalizability of DiVE-k by performing experiments on different LVLMs, such as confirming performance improvements on models like InternVL.

**Questions:**

What would the performance be if the obtained data were used directly for Supervised Fine-Tuning (SFT)? This would help in understanding the advantage of using RL.

---

> ### Author Response · Authors · 2025-11-20
>
> We thank the reviewer for their valuable, positive feedback and recognizing the insights behind our hard-negative mining formulation using top-k predictions, the strong performance gains across tasks, and the clarity of our presentation. Below, we provide our response to address the concerns and questions.
>
> Weaknesses:
>
> 1. There is limited LVLM reasoning based existing work for fine-grained classification. Mentioned in related work are either zero shot (such as FuDD, DesCLIP) or prompt learning based method (such as CoCo, CoCoOp) which uses CLIP model. So it's not fair to compare them with our method. However, We have added CLIP zero shot numbers for completeness. Additionally, we have added SFT numbers (Full training and LoRA training) as baseline.
>
> 2. Thank you for the suggestion. We tested our method on Gemma3-12B model on CUB and flower dataset and present the results in the table below. Similar to QWEN model, DiVE-k consistently improves the harmonic mean (HM) performance on both datasets using Gemma3-12B, achieving gains of +13.07 and +7.83 points over the base model. We have updated our paper to include this results in appendix A.3
> $$
> \\begin{array}{lcccccc}
> \\textbf{Method}
> & \\textbf{CUB} &  &  & \\textbf{Flowers} &  &  \\\\
>  & Base & Novel & HM & Base & Novel & HM \\\\
> \\hline
> \\text{Gemma3-12B}
> & 38.67 & 32.00 & 35.03 & 71.26 & 86.52 & 78.16 \\\\
> \\text{Gemma3-12B*}
> & 46.00 & 37.17 & 41.12 & 69.54 & 86.67 & 77.17 \\\\
> \\text{DiVE-k (Ours)}
> & 58.50 & 40.83 & 48.10 & 84.63 & 87.38 & 85.99 \\\\
> \\Delta \\text{ w.r.t Gemma3}
> & 20.17 & 8.83 & 13.07 & 13.37 & 0.86 & 7.83
> \\end{array}
> $$
>
> Questions:
>
> 1. We have added SFT numbers in Table 1. We observe that Full-SFT yields good accuracy gains on base categories, while its performance deteriorates sharply on novel categories, with an average accuracy drop of 33.5\% relative to the baseline model and a 19.9\% reduction in HM. This sharp degradation highlights SFT’s inability to generalize under the base-to-novel transfer setting.
>
> We hope this clarifies the reviewer’s concerns. We would be happy to answer any further questions or provide additional clarifications if needed. We hope the reviewer will consider raising the score if their concerns have been addressed.

---

> ### Author Response · Authors · 2025-11-25
>
> Dear Reviewer, Thank you again for thoughtful review of our submission. As the end of discussion period is approaching, we are curious to know if our responses to your questions were sufficiently addressed. We welcome any further questions or discussions you may have.

---

### Official Review · Reviewer_G36b · 2025-10-27

**Soundness:** 1
**Presentation:** 3
**Contribution:** 2
**Rating:** 2
**Confidence:** 4

**Summary:**

This paper proposes DiVE-k, a framework to improve fine-grained visual recognition (FGVR) in Large Vision Language Models (LVLMs). The core idea is to address the model's inability to differentiate between visually similar categories. The method first uses an offline step where the base model generates $K$ rollouts to create a top-k set of candidate answers for each image. This set is then used to formulate a Multiple-Choice Question (MCQ). In the second step, the model is trained using Reinforcement Learning (RL) with a simple, verifiable reward (i.e., selecting the correct option letter) to perform differential reasoning among these pre-defined options. The experiments show strong performance gains over baselines like ViRFT and the base QWEN2.5-VL-7B model.

**Strengths:**

The primary strength of this paper is its strong empirical performance. The proposed DiVE-k method achieves significant improvements in base-to-novel generalization, mixed-domain, and few-shot settings, as shown in Tables 1, 2, and 3. The qualitative examples in Figures 4 and 9 are also compelling, illustrating that the model learns to perform more detailed differential reasoning when forced to choose from a set of plausible, similar options, which is a key challenge in FGVR.

**Weaknesses:**

Despite the strong results, I have major concerns about the methodological choices and novelty of this work.
* Limited Novelty: The proposed method's novelty is marginal. At its core, it is a two-step process: 1) an offline data curation step that converts an open-ended generation task into a multiple-choice classification task, and 2) a standard RL training step (GRPO) on this new task. The "differential reasoning" appears to be a direct consequence of this prompt reformatting (from open-ended to MCQ), rather than from a new algorithmic insight.
* Questionable Offline Top-k Generation: The decision to use a static, offline set of top-k options generated by the reference model ($\pi_{ref}$) is a significant weakness. This means the policy ($\pi_\theta$) is trained on a fixed set of problems that were defined by an older version of itself. This introduces a distribution mismatch and severely limits the model's learning. The model is not learning to generate better candidates itself; it's only learning to rank a fixed set of candidates provided to it.Lack of a
* Principled RL Formulation: A more sound and principled approach would be a dynamic, multi-step RL process. For instance, a multi-turn RL agent could first generate its own top-k candidates in a "generation" phase and then, in a "reasoning" phase, select the best one. The entire sequence would then receive a reward. The current method, by decoupling candidate generation (offline) from candidate selection (RL), feels like an ad-hoc pipeline rather than an end-to-end reasoning framework.

**Questions:**

* The core of the method is the offline top-k generation. Why was this choice made over a dynamic, multi-turn RL formulation where the policy first generates its own options and then selects from them within a single episode?
* Could the authors please provide a comparison against a baseline where the options are generated dynamically by the current policy ($\pi_\theta$) at each training step, rather than fixed offline by $\pi_{ref}$?
* Please provide a comparison against a more complete "multi-turn RL" baseline, as described in Weakness #3. This seems like a more correct and challenging setup for this problem.
*Given that the main change is reformatting the problem as an MCQ, how much of the gain is simply from this prompt engineering versus the RL training itself?

**Details Of Ethics Concerns:**

no concerns

---

> ### Author Response · Authors · 2025-11-20
>
> We thank the reviewer for their valuable feedback and recognizing the strong empirical performance of DiVE-k and its ability to encourage fine-grained differential reasoning for FGVR. Below, we provide our response to address the concerns and questions.
>
> Weaknesses:
>
> 1. We respectfully disagree with the characterization that our differential reasoning capability arises solely from prompt reformatting. In Table 1, we report accuracies for both QWEN2.5-VL and ViRFT under the same MCQ-style reformatting (denoted as QWEN2.5* and ViRFT*). The fact that our method substantially outperforms these baselines demonstrates that the gains do not stem from reformatting alone, but from the use of top-k rollouts as options and our offline hard-negative mining strategy, which together enable the model to learn meaningful differential reasoning. Furthermore, as shown in the ablation in Section 4.3.1, simply converting the task into an MCQ format with randomly sampled options yields no improvement, particularly on novel categories. This further supports our claim that the benefits come from the structured top-k option generation and hard-negative selection, rather than from prompt formatting by itself.
>
> 2. We would like to clarify that using an offline top-k option set does not pose the limitation suggested. This step is what enables effective hard-negative retrieval, which we found crucial for efficient and more reliable CoT-style training. While we agree that dynamically generating options with an updated policy could offer closer alignment, the offline options are highly effective in practice: as shown in Fig. 1b, the base model’s Pass@k accuracy is already strong, enabling these candidates to stay well aligned with the model’s underlying distribution.
> Additionally, although the second-stage model is trained to choose the correct answer among the fixed candidates, including the ground-truth option during the offline step exposes the model to a broader set of category distinctions. This is reflected in the consistent improvements in top-k accuracy that we report in the table below. Taken together, these observations indicate that offline top-k generation provides robust and meaningful supervision rather than restricting the model’s learning.
> $$
> \\begin{array}{lcccccccccccc}
> \\textbf{Method}
> & \\textbf{Flowers} &  & \\textbf{CUB} &  & \\textbf{Pets} &  & \\textbf{Cars} &  & \\textbf{Aircraft} &  & \\textbf{Avg} &  \\\\
>  & B & N & B & N & B & N & B & N & B & N & B & N \\\\
> \\hline
> \\text{QWEN2.5-VL}
> & 96.84 & 92.62 & 83.33 & 74.66 & 98.40 & 98.89
> & 86.67 & 90.84 & 89.02 & 90.15 & 90.9 & 89.4 \\\\
> \\text{DiVE-k (ours)}
> & 98.99 & 92.62 & 89.66 & 77.50 & 98.94 & 99.16
> & 89.74 & 90.57 & 92.96 & 90.21 & 94.1 & 90.0 \\\\
> \\Delta
> & 2.15 & 0.00 & 6.33 & 2.84 & 0.54 & 0.27
> & 3.07 & -0.27 & 3.94 & 0.06 & 3.2 & 0.6
> \\end{array}
> $$
>
> 3. We appreciate the suggestion of exploring an end-to-end (e2e) reasoning framework, and we conducted experiments in this direction on the Flowers and the CUB datasets. We evaluated two variants of the proposed idea and reported the results in the table below. In the first variant, we performed e2e training directly on the full training set while in the second variant, we used our offline method to first generate curated hard negative data and then applied e2e training on this set. In the table below, row with e2e refer to end to end suggested training and e2e w/ hn refers to end to end training using hard negative data.
> Overall, these e2e variants did not outperform our proposed method, except for the second variant on Flowers, where we observed a slight improvement on the novel classes when trained using our hard negative examples. We believe that the fully online e2e approach suffers from suboptimal optimization dynamics, sampling top-k candidates from the continuously updated policy causes the model to drive it toward local optima and degrading top-k quality for subsequent batches. In contrast, our offline mining strategy leverages a strong, fixed foundation model to preserve a more global and stable candidate distribution, which in turn enables more reliable optimization and better overall performance.
> $$
> \\begin{array}{lcccccc}
> \\textbf{Method}
> & \\textbf{Flowers} &  &  & \\textbf{CUB} &  &  \\\\
>  & B & N & HM & B & N & HM \\\\
> \\hline
> \\text{Qwen2.5-VL-7B}
> & 84.20 & 83.76 & 83.98 & 63.33 & 48.17 & 54.72 \\\\
> \\text{ViRFT}
> & 84.34 & 84.61 & 84.47 & 65.44 & 51.00 & 57.32 \\\\
> \\text{DiVE-k}
> & \\textbf{97.41} & 88.87 & 92.94 & \\textbf{80.50} & \\textbf{65.50} & \\textbf{72.23} \\\\
> \\text{DiVE-k e2e}
> & 96.55 & 88.79 & 92.51 & 72.50 & 57.50 & 64.13 \\\\
> \\text{DiVE-K e2e w/ hn}
> & 97.13 & \\textbf{89.15} & \\textbf{92.97} & 71.80 & 56.83 & 63.44
> \\end{array}
> $$
>
> Continued in next comment...

---

> > ### Author Response · Authors · 2025-11-20
> >
> > Questions:
> >
> > 1. Answered in Weakness 2 response.
> > 2. Answered in Weakness 3 response.
> > 3. Answered in Weakness 3 response. Also, As we mentioned in Weakness1 response, simple reformatting doesn't lead to improved performance.
> >
> > In our humble opinion, the assigned score does not fully reflect the nature of the concerns raised. We sincerely hope that our responses have helped clarify the highlighted weaknesses and address the reviewer’s questions. Based on this, We kindly request the reviewer to reconsider the score.

---

> ### Author Response · Authors · 2025-11-25
>
> Dear Reviewer, Thank you again for thoughtful review of our submission. As the end of discussion period is approaching, we are curious to know if our responses to your questions were sufficiently addressed. We welcome any further questions or discussions you may have.

---

### Official Review · Reviewer_TDsH · 2025-10-30

**Soundness:** 4
**Presentation:** 3
**Contribution:** 3
**Rating:** 8
**Confidence:** 5

**Summary:**

DiVE-K formulated finegrained classification task into a self-generated MCQ leveraging the base model reasoning ability. This forces the VLM to perform reasoning using a differentiable rewards, leading to better generalization. The paper is well written and the framework is empirically validated, marking it as a strong contribution. Only concern is, there is no/lack of evidences to show the failure scenarios of the proposed solution.

**Strengths:**

- **Formualation**: DiVE-K formulated the task using base model's top-k predictions as a source hard-negative examples to construct the MCQ, and leverages model's reasoning ability with a differentiable reward system, making it a highly effective training method.

- **Differentiable Reasoning**- The MCQ format inherently encourages the model to focus on attribute level discriminative reasoning, which beneficial for semantically similar concepts.

- **Reward**: Simple reward based on MCQ index selection overcomes the existing string matching proposal in previous RL based system.

- **Experiments**: The proposed method is empirically evaluated and ablated under different settings. The performance of the DiVE-k is significantly improved with the proposed mechanisms making it as SOTA of the task.

**Weaknesses:**

- **Senstivity to roll outs and MCQ size**:  The performance of DiVE-k is heavily relies on K (number of rollouts) and m (Size of the final MCQ). While it is stated some processing to keep consistency, but there is no ablation on how variations in K and m affect the quality of the negative set and final performance.

- **Failure cases**: There's systematic analysis/discussions  when this differentiable  reasoning analysis could fail, because this reasoning in next step relies on base model capacity of identifying and including the ground truth in the rollouts. Therefore, it naturally raises question, what is the bottleneck of the proposed solution: initial option mining or differential reasoning chain. For example, in Fig 5, performance on Pets dataset drops when increasing the  top-k generations. Why did that happen?

- **Applicability**: As mentioned earlier, the performance of the base model could influence the final performance. Authors could consider testing the algorithms with different models to demonstrate the proposed solution as model-agnostic.

- **Evaluation**: Figure 8: Provides a prompt intended to evaluate the fine-grained image classification results. Is this used for only property models or all the experiments?  If it is used for open-sourced model as well, what is the reason?. Given the prediction and groundtruth, performance score can be easily evaluated. Beyond accuracy, there's no other evaluation metric considered in the analysis.

**Questions:**

1.  Could authors provide computational cost as it involves multiple step pipeline during training and inference?
2. Could authors include zero-shot performance of the CLIP model on same classes (which can be obtained from original paper)? It will inform how models trained with differ in performance on same dataset?
3. Why objective function is moved to supplementary?
4. Does Table 1 has any results with supervised finetuning?If not, could authors include it as well.
5. I’m unsure if this is a typo or an actual output. In Figure 3, all the reasoning steps of DiVE-k compare “X3” and “X6”, but the final prediction is B (which I assume is the second prediction from the top-k). However, the reasoning step itself contains a statement that it cannot be “X5”.

---

> ### Author Response · Authors · 2025-11-20
>
> We thank the reviewer for their valuable feedback and for recognizing the strengths of our DiVE-k formulation, its MCQ-based differentiable reward, and its ability to encourage fine-grained discriminative reasoning. We also appreciate their acknowledgment of our strong experimental performance across settings. Below, we provide our response to address the concerns and questions.
>
> Weaknesses:
> 1. $\textbf{Senstivity to roll outs and MCQ size:}$ We chose K=20 to ensure a sufficiently diverse option set. Although we do not include a formal ablation on K and m, our experiments indicated that with smaller values of K, the rollouts often repeat the same candidates, resulting in fewer unique options (i.e., a smaller effective m). This leads to sub-optimal learning because distinguishing among very few options (for example, m=2) is a much easier task than reasoning over a richer set such as m=5. As a result, larger K values help maintain diversity in the option pool and support more meaningful differential reasoning during training.
>
> 2. $\textbf{Failure Cases:}$ We appreciate the reviewer’s comment and have updated the paper with an additional ablation section  (4.3.4) addressing this point. We also provide the table below. Our method demonstrates improvements in both option-mining (top-k) accuracy and differential reasoning. However, the primary bottleneck remains in the differential reasoning stage for most of the dataset as Top-k accuracy is high. On Pets, top-k accuracy saturates early (94.16→98.94) see table below, so increasing K mainly adds highly similar distractors that make differential reasoning harder, whereas other dataset such as CUB, rising top-k accuracy (71.83→89.66) still improves true-option inclusion, leading to continued accuracy gains.
> $$
> \\begin{array}{lcccccccccccc}
> \\text{Top-k accuracy (step 1)} \\\\[6pt]
> \\textbf{Method}
> & \\textbf{Flowers} &  & \\textbf{CUB} &  & \\textbf{Pets} &  & \\textbf{Cars} &  & \\textbf{Aircraft} &  & \\textbf{Avg} &  \\\\
>  & B & N & B & N & B & N & B & N & B & N & B & N \\\\
> \\hline
> \\text{QWEN2.5-VL}
> & 96.84 & 92.62 & 83.33 & 74.66 & 98.40 & 98.89
> & 86.67 & 90.84 & 89.02 & 90.15 & 90.9 & 89.4 \\\\
> \\text{DiVE-k (ours)}
> & 98.99 & 92.62 & 89.66 & 77.50 & 98.94 & 99.16
> & 89.74 & 90.57 & 92.96 & 90.21 & 94.1 & 90.0 \\\\
> \\Delta
> & 2.15 & 0.00 & 6.33 & 2.84 &  0.54 & 0.27
> & 3.07 & -0.27 & 3.94 & 0.06 & 3.2 & 0.6
> \\end{array}
> $$
> $$
> \\begin{array}{lcccccccccccc}
> \\text{MCQ accuracy (Step 2)} \\\\[6pt]
> \\textbf{Method}
> & \\textbf{Flowers} &  & \\textbf{CUB} &  & \\textbf{Pets} &  & \\textbf{Cars} &  & \\textbf{Aircraft} &  & \\textbf{Avg} & \\\\
>  & B & N & B & N & B & N & B & N & B & N & B & N \\\\
> \\hline
> \\text{QWEN2.5-VL}
> & 93.65 & 94.90 & 82.20 & 78.62 & 86.99 & 94.85
> & 73.03 & 84.32 & 71.10 & 75.42 & 81.4 & 85.6 \\\\
> \\text{DiVE-k (ours)}
> & 98.40 & 95.98 & 89.78 & 84.52 & 90.05 & 95.00
> & 76.90 & 84.10 & 73.25 & 76.59 & 85.7 & 87.2 \\\\
> \\Delta
> & 4.75 & 1.08 & 7.58 & 5.90 & 3.06 & 0.15
> & 3.87 & -0.22 & 2.15 & 1.17 & 4.3 & 1.6
> \\end{array}
> $$
> $$
> \\begin{array}{l}
> \\text{Top-k accuracy with varying K} \\\\[6pt]
> \\begin{array}{lcc}
> \\textbf{K} & \\textbf{Pet} & \\textbf{CUB} \\\\
> \\hline
> 2  & 94.16 & 71.83 \\\\
> 5  & 95.75 & 80.17 \\\\
> 10 & 97.87 & 85.83 \\\\
> 15 & 98.14 & 88.83 \\\\
> 20 & 98.94 & 89.66
> \\end{array}
> \\end{array}
> $$
>
> 3. $\textbf{Applicability:}$ We thank reviewer for the suggestion. We tested our method on Gemma3-12B model on CUB and flower dataset and present the results in the table below. Similar to QWEN model, DiVE-k consistently improves the harmonic mean (HM) performance on both datasets using Gemma3-12B, achieving gains of +13.07 and +7.83 points over the base model. We have updated our paper to include this results in appendix A.3
> $$
> \\begin{array}{lcccccc}
> \\textbf{Method}
> & \\textbf{CUB} &  &  & \\textbf{Flowers} &  &  \\\\
>  & Base & Novel & HM & Base & Novel & HM \\\\
> \\hline
> \\text{Gemma3-12B}
> & 38.67 & 32.00 & 35.03 & 71.26 & 86.52 & 78.16 \\\\
> \\text{Gemma3-12B*}
> & 46.00 & 37.17 & 41.12 & 69.54 & 86.67 & 77.17 \\\\
> \\text{DiVE-k (Ours)}
> & 58.50 & 40.83 & 48.10 & 84.63 & 87.38 & 85.99 \\\\
> \\Delta \\text{ w.r.t Gemma3}
> & 20.17 & 8.83 & 13.07 & 13.37 & 0.86 & 7.83
> \\end{array}
> $$
>
> 4. $\textbf{Evaluation:}$ We use the same prompt for both open-source and proprietary models. From our perspective, there is no inherent reason why an open-source model would require a different prompt for evaluating fine-grained classification performance, but we may be misunderstanding the reviewer’s concern and would be happy to clarify further.
> Regarding evaluation metrics, we follow prior work in reporting accuracy as the primary measure for fine-grained recognition. That said, we are open to including additional analyses if the reviewer believes specific metrics would provide further insight.
>
> Continued in next comment...

---

> > ### Author Response · Authors · 2025-11-20
> >
> > Questions:
> >
> > 1. Sure. we have included a summary of the computational cost in the table below (for K=20). We have also included it in appendix A.4 of our paper. Though our method adds additional compute, We note that under the same compute budget (specifically, when using K=1, which corresponds to a single-pass greedy decoding identical to existing methods) our approach still outperforms prior works on 4 out of the 5 datasets (Figure 5). This suggests that the benefits of our framework are not solely tied to increased computation. Moreover, when additional computation is allowed, the performance gains become substantial, providing a trade-off between computation and accuracy that can be chosen based on the application requirements.
> > $$
> > \\begin{array}{l}
> > \\textbf{Per-sample inference time comparison} \\\\[6pt]
> > \\begin{array}{lcc}
> > \\textbf{Dataset} & \\textbf{One Step} & \\textbf{Two Steps} \\\\
> > \\hline
> > \\text{Flowers} & 2.50\\,s & 16.02\\,s \\\\
> > \\text{CUB}     & 2.77\\,s & 14.58\\,s \\\\
> > \\text{Pet}     & 2.23\\,s & 8.25\\,s \\\\
> > \\text{Average} & 2.50\\,s & 12.95\\,s
> > \\end{array}
> > \\end{array}
> > $$
> >
> > 2. Thank you for the suggestion. We initially did not include CLIP zero-shot numbers because we believed it might not be a fair comparison to our setting. However, we agree that having these results can provide useful context. Accordingly, we have updated Table 1 to include the CLIP zero-shot performance for completeness.
> >
> > 3. The objective function in our method follows the standard GRPO formulation and are used in subsequent works such as ViRFT. Because it is unchanged and due to space limitations in the main paper, we initially moved the detailed objective to the supplementary material.
> > Now that we have an additional page available, we have moved the objective function into the main section.
> >
> > 4. We appreciate the reviewer’s feedback. We have updated Table 1 to include results for both full supervised fine-tuning (SFT-full) and parameter-efficient fine-tuning (SFT-LoRA). Although Full-SFT yields strong accuracy gains (15.1\%) on base categories, its performance deteriorates sharply on novel categories, with an average accuracy drop of 33.5\% relative to the baseline model and a 19.9\% reduction in HM. This sharp degradation highlights SFT’s inability to generalize under the base-to-novel transfer setting. LoRA SFT doesn’t provide any significant gain.
> >
> > 5. Thank you for pointing this out. The issue was due to a typo in the ordering of the options in the figure. We have updated Figure 3.
> >
> > We hope our response addresses the reviewer’s questions. We would be happy to answer any further questions or provide additional clarifications if needed.

---

> > > ### Comment · Reviewer_TDsH · 2025-11-24
> > >
> > > I appreciate the authors for sharing additional ablations and insights. Most of my concerns have been addressed, so I will maintain my current score.

---

### Official Review · Reviewer_Gbyt · 2025-10-31

**Soundness:** 2
**Presentation:** 3
**Contribution:** 2
**Rating:** 4
**Confidence:** 5

**Summary:**

The paper proposes DiVE-k framwork which uses the top-k generation of base model as a training signal for fine-grained image classification. For each training image, DiVE-k creates a multiple-choice question from the model's top-k outputs and uses RL to train the model to select the correct answer. Experiments on standard base-to-novel generalization and mixed-domain zero-shot base-to-novel generalization demonstrate the effectiveness of the proposed method.

**Strengths:**

1. It transfers the open-world classification task into closed-world classification task, which is a promising way to settle the problem of brittle exact string-match reward.
2. The evaluation metrics is reasonable for fine-grained classification task. Previous work typically uses string matching to evaluate the accuracy, while this paper uses the LLM to determine whether the prediction and ground truth belong to the same fine-grained category or not.

**Weaknesses:**

1. A direct way to construct the hypotheses set is to select the most similar top-k categories by CLIP text features, and the advantage of the proposed offline option mining lacks experimental support.
2. Since the framework uses a two-step pipeline with chain-of-thought, it incurs additional computational cost due to the requirement of two forward passes.
3. The final accuracy heavily depends on the recall of the first inference step, which is not presented in the experimental results.

**Questions:**

1. What is the performance if the ground truth label is already included in the options, i.e., the typical closed-world multiple-choice setting of evaluating LVLM's fine-grained classification performance?
2. What is the performance if the model is trained to obtain options and do differential reasoning in one step?

---

> ### Author Response · Authors · 2025-11-20
>
> We thank the reviewer for their valuable feedback and recognizing the strengths of our work, especially our closed-world reformulation and model-based semantic evaluation. Below, we address their concerns and questions.
>
> Weaknesses:
> 1. We agree that constructing the hypotheses set using the top-k most similar categories from a text-embedding model (e.g., CLIP) is a natural baseline. To evaluate this idea, we included a dedicated comparison in Section 4.3.1 (Table 4). The results show that while the text-embedding–based option construction provides some improvement, our offline option mining strategy leads to substantially stronger performance on both base and novel categories. This empirical evidence supports our motivation: leveraging the model’s own top-k generations yields a more informative and discriminative option set, thereby enabling more effective differential reasoning during training.
>
> 2. We agree that our two-step pipeline introduces additional computational cost due to the need for two forward passes and acknowledge it in our limitations section. However, we note that even under the same compute budget (specifically, when using K=1, which corresponds to a single-pass greedy decoding identical to existing methods) our approach still outperforms prior works on 4 out of the 5 datasets (Figure 5). This suggests that the benefits of our framework are not solely tied to increased computation. Moreover, when additional computation is allowed, the performance gains become substantial, providing a trade-off between computation and accuracy that can be chosen based on the application requirements.
>
> 3. We provide the Pass@k accuracy of the base model in Fig. 1(a), which shows that the base model already exhibits high recall in the first-step generation across all datasets. This high recall directly motivates our offline option–generation strategy, as the correct label is typically present among the model’s top-k candidates.
> Furthermore, we are providing our top-k accuracy in table below. After training, we observe an additional improvement in top-k recall. We have included the corresponding numbers in the revised version of the paper (sec 4.3.4) to more clearly demonstrate how recall evolves before and after RL training.
> $$
> \\begin{array}{lcccccccccccc}
> \\textbf{Method}
> & \\textbf{Flowers} &  & \\textbf{CUB} &  & \\textbf{Pets} &  & \\textbf{Cars} &  & \\textbf{Aircraft} &  & \\textbf{Avg} &  \\\\
>  & B & N & B & N & B & N & B & N & B & N & B & N \\\\
> \\hline
> \\text{QWEN2.5-VL}
> & 96.84 & 92.62 & 83.33 & 74.66 & 98.40 & 98.89
> & 86.67 & 90.84 & 89.02 & 90.15 & 90.9 & 89.4 \\\\
> \\text{DiVE-k (ours)}
> & 98.99 & 92.62 & 89.66 & 77.50 & 98.94 & 99.16
> & 89.74 & 90.57 & 92.96 & 90.21 & 94.1 & 90.0 \\\\
> \\Delta
> & 2.15 & 0.00 & 6.33 & 2.84 & 0.54 & 0.27
> & 3.07 & -0.27 & 3.94 & 0.06 & 3.2 & 0.6
> \\end{array}
> $$
>
> Continued in next comment..

---

> ### Author Response · Authors · 2025-11-20
>
> Questions:
> 1. The performance in the closed-world MCQ setting indeed depends on how the non-ground-truth options are constructed. As shown in Table 4, if the additional options are sampled randomly, it leads to limited learning as the task becomes substantially easier to pick among unrelated options. In contrast, when the options are drawn from the base model’s own K-rollouts, as in our proposed setting, the base model often struggles to identify the correct answer. This demonstrates that top-k candidates produced by the model itself form a much more challenging and informative hypothesis set. We have included a table in the revised version section 4.3.4 reporting the MCQ accuracy of both the base model and the trained model under this setting for completeness.
> $$
> \\begin{array}{lcccccccccccc}
> \\textbf{Method}
> & \\textbf{Flowers} &  & \\textbf{CUB} &  & \\textbf{Pets} &  & \\textbf{Cars} &  & \\textbf{Aircraft} &  & \\textbf{Avg} & \\\\
>  & B & N & B & N & B & N & B & N & B & N & B & N \\\\
> \\hline
> \\text{QWEN2.5-VL}
> & 93.65 & 94.90 & 82.20 & 78.62 & 86.99 & 94.85
> & 73.03 & 84.32 & 71.10 & 75.42 & 81.4 & 85.6 \\\\
> \\text{DiVE-k (ours)}
> & 98.40 & 95.98 & 89.78 & 84.52 & 90.05 & 95.00
> & 76.90 & 84.10 & 73.25 & 76.59 & 85.7 & 87.2 \\\\
> \\Delta
> & 4.75 & 1.08 & 7.58 & 5.90 & 3.06 & 0.15
> & 3.87 & -0.22 & 2.15 & 1.17 & 4.3 & 1.6
> \\end{array}
> $$
>
> 2. While a single-step formulation, where the model both generates options and performs differential reasoning in one forward pass, would ideally reduce the two-pass computational overhead, it is not straightforward to realize in practice. As a simple baseline, we prompt the model to perform these two operations within a single response and report its accuracy on the CUB dataset in table below. We observe a noticeable drop in performance compared to standard CoT-based prompting. Qualitative analysis shows that the model often does not reliably follow the given instructions as it often fails to generate relevant candidate options, sometimes introduces irrelevant categories, and struggles to carry out differential reasoning, leading to increased hallucinations and degraded accuracy.
> $$
> \\begin{array}{lccc}
> \\textbf{Method} & \\textbf{Base} & \\textbf{Novel} & \\textbf{HM} \\\\
> \\hline
> \\text{Qwen2.5-VL-7B}
> & 63.33 & 48.17 & 54.72 \\\\
> \\text{one-step prompting}
> & 58.83 & 42.33 & 49.23
> \\end{array}
> $$
>
> We hope this clarifies our contributions and addresses the reviewer’s concerns. We would be happy to answer any further questions or provide additional clarifications if needed. We sincerely request the reviewer to consider raising the score if their concerns have been addressed.

---

> ### Author Response · Authors · 2025-11-25
>
> Dear reviewer, Thank you again for thoughtful review of our submission. As the end of discussion period is approaching, we are curious to know if our responses to your questions were sufficiently addressed. We welcome any further questions or discussions you may have.

---

### Author Response · Authors · 2025-12-01
**Summary (1/3)**

We thank the reviewers for their constructive feedback. In light of the recent incident and to assist the Area Chair in navigating our rebuttal, we are providing a high-level summary of our revisions and an overview of our responses to the key concerns raised by each reviewer. Full details can be found in our individual comments.

**High Level Summary**

We are encouraged that reviewers found DiVE-k framework highly effective, which converts open-world classification into a closed-world MCQ setting using the model’s own top-k predictions, enabling fine-grained, differentiable reasoning and yielding strong empirical gains across datasets and settings.

(reviewer GByt, TDsH) Ablation on performance of each step of DiVE-k: New ablations show DiVE-k improves both top-k recall and reasoning accuracy. Picking correction options among hypothesis sets remains the bottleneck.

(reviewer TDsH, CNJv) Generalization to other backbone: DiVE-k is model-agnostic: applying it to Gemma3-12B yields substantial gains.

(reviewer G36b) Differential reasoning is not a consequence of prompt reformatting: Simply reformatting as MCQ does not drive improvements—gains come from structured hard-negative mining + differential reasoning.

(reviewer G36b) End to End RL training to generate top-k: End-to-end candidate generation underperforms due to unstable optimization; fixed offline candidates provide more reliable supervision.

(reviewer GByt, TDsH) Extra Computational cost: Extra compute is not the reason for improved performance: even with identical compute (K=1), our method beats prior work. Additional improvements is achieved with extra compute.

(reviewer TDsH, CNJv) More baselines: Added CLIP, SFT, and LoRA SFT baselines; SFT harms novel-class generalization significantly despite strong base-class accuracy.


**Note**: We believe the rating of 2 by Reviewer G36b is not commensurate with the technical issues raised. Having addressed all stated concerns, We trust AC will consider this context when evaluating the consensus.

---

> ### Author Response · Authors · 2025-12-01
> **Summary 2/3**
>
> **Summary of response to individual reviewers**
>
> **Reviewer GByt**
>
> The reviewer’s main concerns centered on whether our option-construction method is empirically justified and two-step pipeline’s additional computation. In our response, we clarified that we included a direct comparison to the text-embedding-based (using gemini embedding model) top-k baseline (Table 4), showing that our offline option-mining strategy yields substantially stronger results and forms a more informative hypothesis set. We also acknowledged the additional compute of the two-step pipeline (with quantitative numbers appendix A.4), but demonstrated that even under identical compute (K=1), our method outperforms prior work on most datasets, confirming that the gains are not merely due to extra compute.
> The reviewer also raised questions about first-step recall, closed-world MCQ performance, and whether the entire procedure could be done in one step. We provided explicit top-k recall numbers (Sec. 4.3.4), showing high initial recall and further improvements after RL training. Closed-world MCQ performance (where groundtruth is always present among options) depends on how the non-ground-truth options are constructed. In our experiment (Table 4) we find that if options are sampled randomly, it leads to limited learning as the task becomes substantially easier to pick among unrelated options. We have included additional ablation in the revised version (section 4.3.4) reporting the closed world MCQ performance and find that closed-world performance improves significantly when trained using our proposed method. Finally, we tested a one-step variant and found a clear performance drop (Table in response) due to unreliable option generation and weaker reasoning, reinforcing the necessity of our two-step design.
>
> **Reviewer TDsH**
>
> The reviewer’s primary concerns were ablation on performance of each step of our proposed method,  applicability of proposed method using other models, computational overhead comparison and more baselines such as CLIP and SFT. For ablation on accuracy of each step, We added a new ablation section (4.3.4) showing that DiVE-k improves both top-k recall and reasoning accuracy, and we explained dataset-specific behaviors such as the Pets performance drop, which arises because its top-k recall saturates early.
>
> To address the model agnostic nature of our method, We applied DiVE-k to a different architecture (Gemma3-12B), observing consistent performance gains (refer to the reviewer’s response or A.3 of the paper for table) of +13.07 and +7.83 points over the QWEN2.5 model for CUB and Flowers datasets respectively. We further provided computation overhead details (appendix A.4) and note that under the same compute budget our approach still outperforms prior works (Figure 5).
> Finally, we included CLIP and SFT baselines (full and LoRA fine-tuning). SFT shows poor generalization compared to our method. Full-SFT yields strong accuracy gains (15.1%) on base categories but its performance deteriorates sharply on novel categories, with an average accuracy drop of 33.5% relative to the baseline model and a 19.9% reduction in HM.
> The reviewer acknowledged that our responses addressed most of their concerns.
>
> **Reviewer G36b**
>
> The reviewer’s main concerns are based on 1) differential reasoning arises as prompt reformatting 2) use of offline top-k option generation and not using end to end training to generate top-k using updated policy. In our rebuttal, we clarified that the gains do not arise from MCQ prompt reformatting alone: Table 1 shows that both QWEN2.5-VL and ViRFT, when evaluated under the same MCQ-style setup, perform substantially worse than DiVE-k. Additional ablations (Section 4.3.1) demonstrate that simply converting the task into MCQ, with random or weak distractors, yields no improvement, confirming that the performance gains come from the structured top-k hard-negative mining and our differential reasoning training, not from prompt reformatting.
>
> We also responded directly to the request for end to end training baselines. We implemented end-to-end variants where the policy dynamically generates its own candidates during training. Across CUB and Flowers, end to end formulation underperformed relative to DiVE-k. We attribute this to unstable optimization when the option set is generated by a continually changing policy, which degrades candidate quality over time. In contrast, the fixed offline candidates generated by a strong foundation model provide a more globally consistent supervision signal. As the base model already has strong Pass@k accuracy, It produces stable, high-quality candidates that enable reliable hard-negative retrieval. The training process further improves top-k recall, indicating that learning is not constrained by the offline set but rather benefits from its stability and diversity.

---

> > ### Author Response · Authors · 2025-12-01
> > **Summary 3/3**
> >
> > **Reviewer CNJv**
> >
> > The reviewer’s main concerns were the limited comparison to related work, lack of diverse backbone models, and comparison to supervised fine-tuning (SFT). In our rebuttal, we clarified that most prior methods discussed in the related work are either zero-shot (without training), such as such as FuDD and DesCLIP or prompt-learning approaches such as CoCo, and CoCoOp based on CLIP, and therefore not directly comparable to LVLM-based reasoning methods like ours. Nonetheless, for completeness, we added CLIP zero-shot results as an additional reference, and we expanded our baselines by including both full SFT and LoRA SFT results.
> >
> > To address the concern about model generalizability, we conducted new experiments with a different LVLM backbone, Gemma3-12B, on the CUB and Flowers datasets. DiVE-k again produced substantial improvements (+13.07 and +7.83 HM), demonstrating consistent gains over Qwen2.5-VL. Finally, we added SFT baselines. While Full-SFT improves base-class accuracy, it catastrophically harms novel-class performance (−33.5% on average), leading to a large HM drop.

---

### Meta-Review · Area_Chair_rXeV · 2026-01-06

**Summary:**

The paper initially received mixed scores: 8, 6, 4, 2. The main concerns include: (1) additional results to verify the advantages of the proposed method; (2) further discussion of failure cases; (3) results with more models/baselines/backbones; (4) clearer clarification of technical details; and (5) limited novelty. The AC has carefully read the reviews and the rebuttal, and finds that the authors have largely addressed these concerns.

Given these considerations, the AC believes the main concerns have been addressed, and the reviewers are likely to revise their scores to be positive (e.g., 8, 6, 6, 6). The AC therefore recommends acceptance.

**Reviewer Concerns:**

Solved Concerns:

* Reviewer Gbyt: Additional results to verify the advantages of the proposed method.
* Reviewer TDsH: (1) Additional results to verify the advantages of the proposed method; (2) further discussion of failure cases; (3) results with more models; (4) clearer clarification of technical details.
* Reviewer G36b: (1) Limited novelty; (2) results with more related baselines; (3) clearer clarification of technical details.
* Reviewer CNJv: Results with more related works and backbones.

**Reviewer Scores:**

The concerns raised by all reviewers were adequately addressed. As a result, Reviewers Gbyt and G36b may raise their scores to a positive score (e.g., 6), while the other two reviewers are likely to maintain their original positive scores (8 and 6).

---

### Decision · Program_Chairs · 2026-01-26

Accept (Poster)